# A Spatial-Frequency Domain Associated Image-Optimization Method for Illumination-Robust Image Matching

**DOI:** 10.3390/s20226489

**Published:** 2020-11-13

**Authors:** Chun Liu, Shoujun Jia, Hangbin Wu, Doudou Zeng, Fanjin Cheng, Shuhang Zhang

**Affiliations:** College of Surveying and Geo-informatics, Tongji University, Shanghai 200092, China; liuchun@tongji.edu.cn (C.L.); 1833538@tongji.edu.cn (S.J.); 4zengdoudou@tongji.edu.cn (D.Z.); cfj@tongji.edu.cn (F.C.); 14zsh@tongji.edu.cn (S.Z.)

**Keywords:** image matching, image optimization, structure from motion, multi-view stereo, spatial and frequency domain analyses

## Abstract

Image matching forms an essential means of data association for computer vision, photogrammetry and remote sensing. The quality of image matching is heavily dependent on image details and naturalness. However, complex illuminations, denoting extreme and changing illuminations, are inevitable in real scenarios, and seriously deteriorate image matching performance due to their significant influence on the image naturalness and details. In this paper, a spatial-frequency domain associated image-optimization method, comprising two main models, is specially designed for improving image matching with complex illuminations. First, an adaptive luminance equalization is implemented in the spatial domain to reduce radiometric variations, instead of removing all illumination components. Second, a frequency domain analysis-based feature-enhancement model is proposed to enhance image features while preserving image naturalness and restraining over-enhancement. The proposed method associates the advantages of the spatial and frequency domain analyses to complete illumination equalization, feature enhancement and naturalness preservation, and thus acquiring the optimized images that are robust to the complex illuminations. More importantly, our method is generic and can be embedded in most image-matching schemes to improve image matching. The proposed method was evaluated on two different datasets and compared with four other state-of-the-art methods. The experimental results indicate that the proposed method outperforms other methods under complex illuminations, in both matching performances and practical applications such as structure from motion and multi-view stereo.

## 1. Introduction

Image matching is used to establish the correspondence between two or more images of the same scene taken from different viewpoints by the same or different sensors. In particular, feature-based image matching methods extract distinctive structures from the images, which depend heavily on image details (i.e., physical structures and textures) and naturalness as salient features for image matching [1,2,3,4,5,6]. The technique possesses the merits of high computational efficiency, high theoretical accuracy, and insensitivity to geometric deformations and differences [7]. As a result, feature-based image matching has received a lot of attention in the field of computer vision [8,9,10,11], photogrammetry and remote sensing [12,13,14,15], in applications [16,17,18] such as multiple view 3D reconstruction, remote sensing image fusion and visual localization.

The intensity of an image is the result of visual sensors, which is a combination of the scene information and the illumination situation. More specifically, image component is viewed as a product of irradiance and reflectance [19]. The reflectance is decided by the spectral property of the imaging scene, which is mostly related to image details [20]. The irradiance is essentially the received illumination at each point of the imaging scene, which represents the image naturalness [21]. However, complex illuminations can be easily induced by light sources, vignetting, exposure differences and other factors [22], which are inevitable in practical situations. In this paper, the complex illuminations are generally categorized into two classes. The first is defined as extreme illuminations, which includes very weak and bright illuminations. In this case, the images may not provide enough distinctive details for feature extracting and matching owing to the inadequate irradiance in weak illumination and the excessive irradiance in bright illumination [23]. The second class is the changing illumination, in which the overlap regions of images with the same reflectance present obvious differences in the irradiance. Under this condition, color of objects and their textures might present significant variations [24]. Consequently, the complex illuminations significantly influence the image naturalness and details, and thus seriously degrade the performance of the image matching methods.

A variety of techniques have been proposed to address the problem of the complex illuminations in image matching. Numerous feature detectors or descriptors are available in the literature [25,26,27,28,29]. The majority of these methods can be used for illumination-invariant image matching, but show low applicability in practical situations [30]. In addition, most existing image-optimization methods [22,31] were applied to remove illumination variations for visual perception, rather than improving image matching. In this paper, a spatial-frequency domain associated image-optimization method, which optimizes the image naturalness and details, is particularly proposed to improve image matching in the presence of the complex illuminations.

## 2. Related Work

### 2.1. Illumination-Robust Feature-Based Matching

Various radiation-invariant feature-matching methods have recently been proposed [32], including order-based descriptors, phase congruency (PC)-based algorithms, and self-similarity-based methods. Order-based methods that rely on the relative ordering of pixel values are invariant to most radiometric changes. The census transform was introduced to compute the visual correspondence based on intensity comparisons [33,34]. For tolerance to illumination variation, the local binary pattern operator and its extensions have been investigated [35] to construct order-based features for each pixel by comparing the intensity of pixels with that of adjacent pixels. The ordinal spatial intensity distribution [36] was constructed by ordering the pixel intensities in both the ordinal and spatial spaces. To obtain less sensitivity to illumination differences, PC-based descriptors have also been proposed. For example, the histogram of orientated phase congruency descriptor [14] was constructed for toleration to the illumination distribution of images based on the physical structure contained in images. Similarly, the PC-based structural descriptor [13] was built based on a PC structural image to obtain less sensitivity to nonlinear variations in illumination. Another recent approach based on local self-similarity (LSS) has been used as an illumination-invariant descriptor. For example, a brightness description method [37] was introduced for quantitatively describing radiation changes based on the geometric moment of the neighboring pixels. The dense LSS [38] was developed based on the robustness of the shape similarity for image correlation under changing illumination. These methods, which seek to develop novel feature-matching algorithms, achieve superior performance in illumination-invariant image matching. However, they may show poor performance under extreme illumination condition, and could not be used in the practical applications when images contain both geometric and radiometric variations.

### 2.2. Image Optimization in the Spatial Domain

Image optimization in the spatial domain can be realized based on the color constancy, histogram equalization and sharpness (HES) and illumination map estimation. As with color constancy [39], the edge-based color constancy (EBCC) algorithm was proposed [40] based on a new gray-edge hypothesis, which is achromatic for the average edge diversity in a scene. The EBCC algorithm was improved [41] using an edge weighting scheme with the use of distinct edge types. In [42], a stereo color histogram equalization method was proposed, which allied accurate disparity maps and color-consistent images to process stereo images with changing illumination. In addition, the HES approaches have been widely employed to correct the brightness difference. A histogram equalization-based algorithm [43] was particularly designed to improve the quality of stereo image matching with illumination differences. The adaptive histogram equalization technique in [44] was proposed to improve weak illumination and low contrast contained in matched images. A modified HE-based contrast enhancement technique was designed to refine the histograms into sub-histograms, which enhances the contrast for non-uniform illuminated images [45]. Moreover, illumination estimation was used to optimize the images with illumination differences [46]. Low-light image enhancement (LIME) was proposed to enhance a low-light image, which only estimates the illumination map to shrink the solution space and reduce the computational cost [47]. Above all, these techniques can effectively reduce illumination variations while preserving illumination components. However, it is not easy to solve the non-uniform illumination and over-enhancement under the complex illuminations. In addition, these approaches mainly concentrate on the consistency of the color and brightness without an effective enhancement for image features.

### 2.3. Image Optimization in the Frequency Domain

The effect of complex illuminations on image matching can also be alleviated in the frequency domain through PC-based image representations. Based on local frequency analysis (LFA) [22], the texture and edge preserving mappings were computed to preserve the texture information and fine edges of the images, respectively. The two mappings were then combined to obtain an illumination-invariant image presentation. To develop a multisource image registration method, Li and Man [48] proposed the phase correlation of principal phase congruency based on the magnitude of the PC. This technique was further developed [49], where both the orientation and magnitude of the PC were integrated to construct a structural radiation-invariant image correlation method. PC-based methods achieve a high performance for radiation consistency by removing the illumination, but lose the naturalness of the image. This may induce a decrease in the distinction of image features, and hence leads to more incorrect matches. Alternatively, to preserve naturalness of the image, the enhancement algorithm (NPEA) was constructed [50] to equalize the non-uniform illumination, decomposing an image into reflectance and illumination, and then making a balance between image details and illumination. The aim of the NPEA approach is to optimize non-uniform illumination image for perceptual quality, rather than an effective feature enhancement for image matching under the complex illuminations.

Based on the above brief analysis, the issues resolved in this study for the problem of illumination-robust image matching can be summarized as follows:(1)How can we find an effective approach to reduce the radiometric variations while preserving the naturalness for the images under complex illuminations?(2)How can the approach handle inconspicuous image features caused by extreme illuminations without over-enhancement and loss of the naturalness?(3)How can the method apply in practical applications to achieve robust image matching when image sequences contain both geometric and radiometric variations?

## 3. Contribution

In this paper, a spatial-frequency domain associated image-optimization method is specifically designed to improve the accuracy and efficiency of image matching under complex illuminations, by reducing radiometric variations and enhancing image details. Generally, spatial domain analysis can effectively equalize image illumination components. In addition, image illumination and details can be quantitatively represented in the frequency domain. We explicitly take both the spatial domain and frequency domain analyses into account, and propose a novel image-optimization method. More specifically, two simple yet effective models, a spatial domain analysis-based adaptive luminance equalization model and a frequency domain analysis-based feature enhancement model, are developed to reduce the radiometric variations and enhance the image details, respectively. More importantly, instead of removing all illumination components, the proposed method equalizes the illumination component in both spatial and frequency domains in order to restrain over-enhancement and preserve image naturalness. Therefore, the optimized images, which are robust to complex illuminations, can be used with most image-matching algorithms to achieve accurate and efficient image matching. More precisely, the main contributions of this paper are summarized as follows:(1)An adaptive luminance equalization model is proposed based on the spatial domain analysis to equalize non-uniform illumination with preserving the image naturalness.(2)A frequency domain analysis-based feature enhancement model is constructed to enhance the image details without over-enhancement and destruction of naturalness.(3)A spatial-frequency domain associated image-optimization method is proposed by combining the advantages of the spatial and frequency domain analyses to improve image matching in complex illuminations. The demo of our approach can be available at: https://github.com/jiashoujun/image-optimization-for-image-matching.(4)Comprehensive performance evaluation and analysis of the proposed method and four other state-of-the-art methods are presented using real scenario and standard datasets.

## 4. The Proposed Method

This study is aimed at providing optimized images that contain distinct details and naturalness for improving image matching under complex illuminations. For this purpose, a spatial-frequency domain associated image-optimization method is constructed to complete illumination equalization, feature enhancement and naturalness preservation. The flowchart of the proposed method is shown in Figure 1. The input of our algorithm is the original images containing complex illuminations. The adaptive luminance equalization model is performed in the spatial domain to reduce radiometric variations without removing all illumination information. Then, the feature-enhancement model is implemented in the frequency domain to enhance the image features while restraining over-enhancement and destruction of naturalness. The outputs are the optimized images that are resistive to the complex illuminations, and then can be embedded into image-matching schemes to achieve illumination-robust image matching.

### 4.1. Adaptive Luminance Equalization in the Spatial Domain

Image features are severely dependent on image naturalness and details that are expressed in a gray form in the spatial domain [20]. It has been reported that radiometric differences between images caused by complex illuminations are inevitable in real applications. It seriously influences the pixel intensity and image naturalness and details, and degrades the performance of most image matching methods [23,24]. Therefore, we propose an adaptive method to equalize the differences in pixel intensities caused by the complex illuminations, and thus reduce the effect of the complex illuminations on image features, which is the main mission in this section.

On the basis of our review, spatial domain analysis has been demonstrated to be effective to reduce illumination variation and enhance image contrast [42,43,47]. In addition, the spatial domain analysis-based image optimization focus on illumination equalization rather than removing all illumination components, which can preserve more image information than other methods such as phase congruency [22,49]. In view of the success of these methods, instead of removing all illuminations, the paper adopts spatial domain analysis to equalize the pixel intensity of images under the complex illuminations for reducing most radiometric variations. The process of illumination equalization is as shown in Figure 2.

#### 4.1.1. Luminance Intensity Estimation

It has been reported that image component consists of illumination and reflectance components. In addition, the low-pass filter can preserve illumination information and remove reflectance component [51]. Thus, to determine the luminance intensities of input images, the low-pass filter is employed to obtain the illumination maps. The luminance intensity estimation can be expressed as follows: (1)L(x,y)={low(x,y)∗G(x,y)}
(2)$$low\left(x,y\right)={\left[1+c*{\left(\frac{D\left(x,\;y\right)}{D_{0}}\right)}^{2p}\right]}_{\left(x,y\right)\epsilon R^{2}}^{-1}$$
where L(x,y) is the estimated luminance intensity, G(x,y) is the intensity of original image, D(x,y) is the Euclidean distance from the current point to the center of image, c is a shape adjustment parameter, and $D_{0}$ is the low-frequency range. The illumination maps of different illumination intensities are illustrated in Figure 1.

#### 4.1.2. Luminance Equalization Model

Using the estimated illumination map, the adaptive illumination equalization model is constructed in the spatial frequency. The model is used to enhance the brightness of low-light images and weaken the luminance of images under bright light, and correct the luminance values of the corresponding pixels to be on the same level. The luminance equalization model can be expressed as follows: (3)$$F=equalize\;\left\{G\right\}={\left\{255*{\left(\frac{G\left(x,y\right)}{255}\right)}^{\gamma }\right\}}_{x=1,2,\ldots ,m;y=1,2,\ldots ,n}$$
where F is the transformed image, G is the original image, γ is an equalization parameter containing the estimated luminance intensity, and m×n denotes the size of image. The illustration of luminance equalization model is as shown in Figure 2d.

#### 4.1.3. Adaptive Equalizing Scheme

Under the complex illumination, the brightness level may be significantly different between images. Moreover, there is also a variant intensity in an image, due to the non-uniform illumination, as shown in Figure 2a,b. Consequently, a global illumination equalization with constant equalization parameter cannot complete the satisfactory performance for images in the complex illuminations [52]. Hence, in order to adaptively control the equalization parameter γ, a sliding window is used to compute local mean luminance value of the illumination map. The adaptive luminance equalization is then achieved, which is expressed as follows: (4)$$\gamma =\exp\left(ln\left(\varphi \right)*\frac{128-\bar{X}}{128}\right)$$
(5)$$\bar{X}=\frac{\sum_{i=1}^{d}\sum_{j=1}^{d}L\left\{x\left(i,\;\;d\right),\;\;y\left(j,\;\;d\right)\right\}}{d*d}$$
(6)$$\varphi =\{\begin{array}{cc} 0.6 & \sigma <40\\ -0.005\sigma +0.8 & 40\leq \sigma \leq 80\\ 0.4 & \sigma >80 \end{array}$$
where L{x(i, d), y(j, d)} is luminance intensity of illumination map within a sliding window, $\bar{X}$ denotes the mean luminance value within the sliding window, d is the size of sliding window, and φ expresses the adjustment parameter containing the standard deviation (i.e., σ) of the original image. The adaptive equalizing scheme is illustrated as shown in Figure 2c.

Figure 2d illustrates how the luminance values of transformed images change when varying the luminance values and mean luminance value of sliding window. The performance of the adaptive luminance equalization model is shown in Figure 2e,f. Here, a rather low mean luminance value indicates that the image is under weak illumination and equalization parameter γ needs to be decreased. In contrast, a relatively large mean luminance value suggests that the image is under bright illumination. In this case, the equalization parameter γ is adaptively increased.

### 4.2. Frequency Domain Analysis-Based Feature Enhancement

Although most illumination variations can be reduced in the spatial frequency, the structures of objects and their textures remain inconspicuous in the images due to complex illuminations, which also create challenges with respect to feature extraction and matching [23,24]. In this section, an effective feature-enhancement model is proposed to improve the performance of image matching.

As stated above, image component is the product of irradiance and reflectance, which are essentially the illumination and property of imaging scene, respectively [20]. Generally, the irradiance represents the global naturalness, and the reflectance represents local details of an image [50]. The image feature, which is dependent of the image details and naturalness, can be enhanced by heightening the reflectance and restraining the irradiance. In the spatial domain, image enhancement synchronously changes the two components, and hence, these algorithms may not complete the requested performance due to strengthening the irradiance. Moreover, in the frequency domain, Fourier transform (FT)-based frequency component has been used to represent the irradiance and reflectance in [53,54,55,56]. In addition, the irradiance and reflectance are related to low-frequency and high-frequency components in the FT-based frequency spectrum, respectively [19,20]. In view of the theories described above, the irradiance and reflectance components are separated, and image analysis is transformed from the spatial domain to the frequency domain in our strategy. Therefore, a frequency domain analysis-based feature enhancement approach is proposed to enhance image details, while preserving naturalness and restraining over-enhancement. The aim is to enhance inconspicuous image features, and hence improving feature extracting and matching under the complex illuminations.

#### 4.2.1. Irradiance-Reflectance Component Decomposing

Image component is the product of irradiance and reflectance. Starting with the output image of adaptive luminance equalization in the spatial domain F, the irradiance and reflectance components can be separated based on a logarithmic transformation, which are expressed as follows [57]: (7)$$F=\left\{\prod_{i,r}F\left(x,y\right)|\left(x,y\right)\epsilon \left[m,n\right]\right\}=\left\{i\left(x,y\right)*r\left(x,y\right)\right\}$$
(8)$$f=\ln\left\{F\right\}=\{\sum_{i,r}lnF\left(x,y\right)|\left(x,y\right)\epsilon \left[m,n\right]\}$$
where f denotes the transformed image, and i(x,y) and r(x,y) denote the irradiance and reflectance components, respectively. Moreover, fast Fourier transform (FFT) is used to complete the spatial to frequency domain transformation, which can be expressed as follows: (9)$$\mathrm{FFT}\to {\left\{f\left(x,y\right)\right\}}_{x=1,2,\ldots ,m;\;y=1,2,\ldots ,n}$$
(10)$$Z=\left\{\sum_{I,R}Z\left(x,y\right)|\left(x,y\right)\epsilon \left[M,N\right]\right\}=\left\{I\left(x,y\right)+R\left(x,y\right)\right\}$$
where Z is the image component after FFT, I(x,y) and R(x,y) denote the results after FFT of ln(i) and ln(r), respectively.

#### 4.2.2. Multi-Interval Frequency Domain Equalization

It has been demonstrated that removing all illuminations can obviously enhance image details, while the remaining illumination information is useful for preserving image naturalness and restraining over-enhancement [50]. To enhance the image details without losing the naturalness and over-enhancement, a multi-interval frequency domain dividing strategy is designed in our method. More specifically, the image frequency range is divided into low $\left[0,\;D_{1}\right]$, high $\left[D_{2},+\infty \right]$, and middle $\left(D_{1},D_{2}\right)$ frequencies in our strategy, as shown in Figure 3a,b. The low and high frequencies are relevant to the irradiance and reflectance, respectively, and the middle frequency is regarded as the transition between low and high frequencies. Based on the multi-interval frequency domains, a multi-interval frequency domain equalization model is proposed in our strategy, as shown in Figure 3c,d. We restrain the low-frequency component rather than removing all low-frequency information in order to reduce the effect of redundant illumination on image features and preserve the naturalness of the image. In addition, high-frequency component is heightened to enhance image details. Please note that remaining middle-frequency component constant is aimed at restraining over-compression of the illumination and over-enhancement of image details. The multi-interval frequency domain equalization model can be presented as: (11)$$H\left(x,y\right)={\left\{1-\frac{l}{\left[1+c*{\left(\frac{D\left(x,\;y\right)}{D_{1}}\right)}^{n}\right]}+\frac{h}{\left[1+c*{\left(\frac{D_{2}}{D\left(x,\;y\right)}\right)}^{n}\right]}\right\}}_{\left(x,y\right)\epsilon W\times H}$$
where D(x,y) is the Euclidean distance from the current point to the center, c is an adjustment parameter, n is the iteration order, W×H denotes the whole image, and l, h are the compression and enhancement coefficients, respectively.

The frequency domain amplitude distribution is used to determine the range of the frequency domain, which can be expressed as follows: (12)$$A=abs\left(Z\right)=\sqrt{I^{2}\left(x,y\right)+R^{2}\left(x,y\right)}$$
where A denotes the frequency domain amplitude. In Figure 4a, the amplitudes are distributed in a cross shape, and the core of the frequency domain lies in the spatial center of the images, which decreases from the center to the periphery. The low and high frequencies occur at the center and external areas, respectively, where the low-frequency amplitude occurs at the peak, whereas the high frequency has the minimum amplitude. Figure 4a shows that the peak in the low-frequency region is significantly high, whereas the high-frequency region is relatively flat. Thus, in Figure 4b,c, the boundary of the high (i.e., $D_{2}$) and low (i.e., $D_{1}$) frequencies can be obtained based on amplitude distribution. The region of the middle frequency can then be uniquely determined from $D_{1}$ to $D_{2}$.

Subsequently, based on the multi-interval frequency domain equalization model H(x,y), the frequency domain equalization is implemented, which is expressed as: (13)$$E=\left\{Z\left(x,y\right)⨂H\left(x,y\right)\right\}{\;}_{x=1,2,\ldots ,M;\;y=1,2,\ldots ,N}$$
where E presents the equalized image, and operator ⨂ denotes dot product of matrices. Hence, the high-frequency component is strengthened, the low-frequency component is compressed, and the middle frequency can remain constant.

#### 4.2.3. Synthesis of Restrained Irradiance and Heightened Reflectance

As mentioned above, the irradiance and reflectance are related to low-frequency and high-frequency components in the frequency domain, respectively. To achieve reflectance enhancement and irradiance restraint, the inverse fast Fourier transform (IFFT) is employed for frequency-to-spatial domain transformation. In addition, the restrained irradiance and heightened reflectance were synthesized to get the final optimized image. The processes can be represented as follows: (14)$$\mathrm{IFFT}\to {\left\{E\left(x,y\right)\right\}}_{x=1,2,\ldots ,M;\;y=1,2,\ldots ,N}$$
(15)$$Z^{\prime }={\left\{\sum_{I^{\prime },R^{\prime }}Z^{\prime }\left(x,y\right)\right\}}_{\left(x,y\right)\epsilon \left[m,n\right]}=\left\{I^{\prime }\left(x,y\right)+R^{\prime }\left(x,y\right)\right\}$$
(16)$$F^{\prime }=\exp\left(Z^{\prime }\right)={\left\{i^{\prime }\left(x,y\right)*r^{\prime }\left(x,y\right)\right\}}_{x=1,2,\ldots ,m;\;y=1,2,\ldots ,n}$$
where $Z^{\prime }$ is the image component after IFFT, $F^{\prime }$ is the optimized image, and $i^{\prime }\left(x,y\right)$ and $r^{\prime }\left(x,y\right)$ denote the restrained irradiance and heightened reflectance.

Therefore, the proposed algorithm can enhance image details while preserving naturalness to improve image matching under complex illuminations. Please note that for color images, this algorithm is implemented in multiple channels after RGB transformation to preserve the color information.

## 5. Experimental Results and Analysis

To verify the proposed method, this paper conducted image optimization experiments on two different datasets. The matching performances were assessed both visually and numerically. The results of structure from motion (SFM) and multi-view stereo (MVS) were shown to evaluate the applicability of our method. Moreover, four other state-of-the-art approaches, including HES [43], LFA [22], LIME [47] and NPEA [50], were used for comparisons.

### 5.1. Experimental Dataset and Implementation

The proposed method was tested on two datasets, considering extreme and changing illumination. The first dataset contained 623 image pairs taken in a long corridor with extreme and uneven illumination. The scenario covers an area of 2.5 m × 51.8 m, in which four ground control points and six checkpoints were available using the total station in this experiment. All images are collected in RGB format and have a resolution of 2592 × 1728, using Canon 600D SLR cameras. Furthermore, the images cover some planar objects, including walls, floors, and roofs. Thus, the dataset can provide image pairs with the extreme illumination. The other dataset was a standard dataset for an outdoor scenario under changing illumination, from the Department of Engineering Science, University of Oxford. The image pairs are from the public datasets downloaded from the Internet and used in the literature [2,58,59]. The dataset comprises the RGB images with a resolution of 921 × 614. The images contain same scenes under varying illumination. Thus, this dataset can be used for validating our method in changing illumination.

Based on the two datasets, the results obtained with the proposed methods were compared with those of the other image-optimization methods. The other methods, including HES [43], LFA [22], LIME [47] and NPEA [50], which can be classified into two types, namely, spatial and frequency domain analysis. HES is an important family among the spatial domain analysis-based image-optimization approaches and has been proposed for illumination-robust image matching. LIME is also an efficient and standard method to enhance a low-light image based on spatial domain analysis. Additionally, LFA and NPEA are two popular frequency domain analysis-based methods. LFA has been reported to achieve high performance in illumination-invariant image matching. NPEA can equalize non-uniform illumination contained in images while preserving the naturalness. The implementations of these four methods basically accorded with the code and original papers. Our method has two groups of important parameters: frequency domain division (i.e., $D_{1},D_{2}$) and the compression and enhancement coefficients (i.e., l, h). The parameters setting in this paper as shown in Table 1.

### 5.2. Evaluation Criteria

To evaluate the performance of the proposed methods, qualitative and numerical matching criteria were introduced following previous important works [51,60]. The number of features (F) and the number of matches (M) were obtained before using Random Sample Consensus (RANSAC) filtering [61]. The number of precise matches (NPM) was computed using RANSAC with a threshold of [0,5], for which the matches are retained. The processing time (T) was measured using a computer with an Intel single Core i7-8700K CPU and 32 GB of RAM. Moreover, three relative indicators are introduced to neutralize the effect of absolute factors. The first is the matching precision (MP), which is expressed as MP=NPM/M. The second is the ratio of precise matches to the number of features (RPMF), which is the ratio between NPM and the minimum number of features. RPMF can be written as $\mathrm{RPMF}=\mathrm{NPM}/\min\left\{F_{1},\;F_{2}\right\}$, where $F_{1},\;F_{2}$ denote the number of features of left and right images used for image matching, respectively. The ratio of precise matches to the processing time (RPMT) is regarded as another relative indicator to exhibit the computational efficiency, which can be written as RPMT=NPM/T. For NPM, MP, RPMF and RPMT, the higher value means the better performance.

In addition, the image pairs are neither accurate camera matrices nor fundamental matrices, but the accurate homography matrices are available in the Oxford dataset. Consequently, the matching error (ME) is designed to exhibit the accuracy of the image matching. Let $\left(p_{i},\;q_{i}\right)$ be the match after filtering using RANSAC. The ME is defined as follows: (17)$${\mathrm{ME}}_{i}=\sqrt{d{\left(q_{i},\;q_{i}{}^{\prime }\right)}^{2}}$$
where $\left(p_{i},\;q_{i}{}^{\prime }\right)$ is a match calculated through known homography matrices between images. The term $d\left(q_{i},\;q_{i}{}^{\prime }\right)$ presents the Euclidean distance between $q_{i}$ and $q_{i}{}^{\prime }$. 

Using these two datasets, the matching experiments are designed as shown in Figure 5. The first three image pairs from the first dataset are under normal and extreme illuminations (bright and weak). The others are from the Oxford dataset. The fourth image pair is characterized as same scene with changing illumination. The fifth is similar to the fourth but has a decrease in illumination intensity. To determine matching metrics, the image matching was implemented using SIFT and SURF. These two algorithms have promising capabilities for resistance to geometric differences, and can avoid the influence of geometric differences on the matching experiments. In addition, these two algorithms have been significantly applied in computer vision, photogrammetry and remote sensing applications, which are representative among most image-matching methods [25,26].

Moreover, the qualitative and numerical metrics of 3D reconstruction were used to evaluate the effectiveness and applicability of the proposed method. For SFM, the success rate of image localization (SRIL) and the mean reprojection error (RE) are calculated to evaluate the effect of sparse reconstruction, where the SRIL is defined as SRIL=(NSL/NL)×100%; here, NSL is the number of successful image localizations and NL is the number of images calculated. Moreover, to assess the precision of the 3D reconstruction, the total error (TE) for each checkpoint is computed, which is the square sum of errors in X, Y and Z direction. The higher value of SRIL means the better performance. For RE and TE, lower values mean better performance. To analyze the effect of illumination on the precision, four ground control points and six checkpoints under different illumination were selected. The commercial software (Pix4D) was used for carrying out SFM and MVS.

### 5.3. Matching Performance

#### 5.3.1. Performance of Visual Indicators

This section presents the results of image optimization for our method and other techniques as well as their visual matching performances under complex illuminations. As shown in Figure 6 and Figure 7, the blue and red lines represent precise and false matches, respectively. The transformed images with our method contain more uniform brightness, and clearer physical structures and textures. However, the HES approach enhances the edge information of images, but further worsens the uneven distribution of illumination intensity. In contrast to HES, the LFA achieves fine results in illumination invariance at the cost of the naturalness of the image. The LIME and NPEA significantly improve the visualization of images in weak illumination, but provide unsatisfactory results in equalizing bright illumination.

The matching performances are depicted visually in Figure 6 and Figure 7. It is clear from Figure 6 (first image pair) that precise matches of the original images using SIFT in weak illumination are only concentrated on the floor with very few matches on the wall. In addition, the proposed method is more effective than the other methods to increase matches on the wall. As for the result using SURF, the advantage of proposed algorithm is more significant. For the second and third image pairs, the matches using SIFT are primarily distributed on the wall under normal and bright illuminations. These five algorithms significantly increase the number of matches on the wall and floor, and our method outperforms the others in terms of precise matches. The matching results using SURF, as shown in Figure 7, are similar to those using SIFT. In terms of the matching performance under changing illuminations (fourth and fifth image pairs), the most precise matches for the original image are located on the building façade and non-planar objects, as shown in Figure 6 and Figure 7. Other approaches, including HES, LFA and NPEA, increase the precise matches only on the building façade and non-planar objects. For the LIME, the optimized images contain more matches but significantly false matches located in the non-planar objects. However, our method yields better performances in the number and distribution of precise matches. Moreover, the results of SIFT and SURF are similar. Therefore, our method can be considered satisfactory for improving illumination-robust image matching.

#### 5.3.2. Performance of Numerical Indicators

Using the evaluation criteria, the matching performances of our method and other methods were numerically compared and evaluated, as shown in Table 2, Table 3, Table 4 and Table 5 and in Figure 8 and Figure 9. For the first image pair, Table 2 and Table 3 show that the NPM and MP are small, except for our algorithm. This is because our method effectively enhances image features. The LFA provide many false matches owing to losing the image naturalness. The LIME and NPEA significantly increase the number of matches but result in lower matching precision than our approach. Moreover, the relatively large RPMF and RPMT of our strategy indicate high effectiveness and efficiency. Thus, the results suggest that our algorithm can significantly improve image matching under weak illumination even when other techniques fail.

As for the second image pair, the results in Table 2 and Table 3 show that our algorithm and other approaches are helpful for improving the image matching. In particular, the NPM of LFA is higher than that of our method when using the SIFT. This is owing to the advantage of combining texture preserving and edge preserving in LFA. However, our approach achieves higher MP and RPMF compared with those of other methods. Furthermore, our method significantly outperforms the others in efficiency due to the larger RPMT. Thus, our strategy is demonstrated to be more effective for improving image matching under relatively uniform illumination than traditional methods.

For the third image pair, the results are similar to the second image pair. In particular, our method achieves an MP of 0.735 with an NPM of 100, whereas the LFA achieves an MP of 0.374 with an NMP of 120, as shown in Table 2. Using SIFT and SURF, the MP of LFA is lower than that of the original image. Furthermore, the performances associated with the HES, LIME and NPEA seem to be worse in bright illumination conditions because the M and NPM are significantly lower compared with those of our techniques, even original images. Therefore, the results in this experiment indicate the capability of our strategy to improve image matching under bright illumination.

The experiments on the fourth and fifth image pairs are aimed at testing our method under changing illuminations. The results of the fourth image pair show that our method outperforms the others, because the NMP and MP of our approach are larger than that of other methods. As expected, the results of RPMF and RPMT, as indicated in Table 4, reveal a higher effectiveness of our algorithm, expect for the better results of the LFA and NPEA when using SIFT. Moreover, for matching errors, the proposed method provides more accurate matches, because the mean matching errors of our approach are the smallest, except for the result of the LIME. The upper bound of the matching errors can be described with their standard deviation, which is relatively lower in our approach compared with that of the other approaches and original images when using SIFT. For the fifth image pair, the experimental results follow the same pattern as the fourth image pair, and our algorithm achieves better performance than that in the fourth image pair. For example, NMP and RPMT are greater for the fifth image pair, although the matching accuracy is lower when applying SURF owing to the greater illumination variance. Hence, the experimental results show that the proposed method outperforms others under the changing illuminations.

Thus, based on the analysis above, the results demonstrate the better efficiency of the proposed approach for improving the image matching under extreme illumination and higher robustness to illumination variation, compared with other methods.

### 5.4. Application to SFM and MVS

The results of SFM for the first dataset are presented in Table 6 and Table 7. The results indicate that LFA achieves the highest SRIL and RE, which means that LFA can improve the continuity of the 3D model, but decreases the precision of the 3D model. Moreover, LIME and NPEA also significantly reduce reconstruction precision owing to the few precise matches and high RE. Our approach, which contains relatively high SRIL and the lowest RE, achieves a better performance compared with other algorithms. This is because of the better matching performance, which can be proved through the matching performance in Table 2 and Table 3. In terms of precision of 3D model, the results in Table 7 indicate the poor results of other approaches. For example, other methods significantly increase the errors of 3D model, even result in loss of some checkpoints when using HES and LIME, as the “loss” in Table 7. Our method obviously reduces the total error of the 3D model, especially under bright and weak illuminations, and the mean error is as low as 20 mm.

Furthermore, the dense reconstruction performances of MVS are visually depicted in Figure 10. It is clear that the 3D models of the original images and transformed images with other algorithms are incomplete within the red circle, with indistinct walls and floors as well as missing doors. Especially for LFA and NPEA, the results display very little textures and physical structures in the reconstructed model. The 3D reconstruction model with the proposed approach contains more integrated physical structures and richer textural details, as shown in Figure 10f. Thus, the proposed method can significantly improve the integrity and visuality of the 3D dense model, and outperforms the other state-of-the-art approaches.

Therefore, the proposed approach can present higher applicability in both SFM and MVS under complex illuminations compared with other approaches. More importantly, it can be clear that the proposed algorithm can improve image matching to effectively apply in the sparse and dense reconstruction of scenarios under the complex illuminations.

## 6. Discussion

### 6.1. Image Naturalness Assessment

Image naturalness is essential for image optimization to achieve pleasing perceptual quality [3,4,5], which influences the performance of image matching. The spatial-frequency domain associated image-optimization method is proposed to enhance image details while preserving the image naturalness. To quantitatively assess the naturalness preservation performances of the image-optimization methods, we used a quantitative measure, namely, the lightness-order-error (LOE), following previous work [50].

The quantitative measurement results of LOE is shown in Table 8. The results demonstrate that the proposed method achieves a moderate level of the naturalness preservation performance, outperforming LFA and LIME. This reason could be that LFA and LIME restrain illumination components contained in the images and lose naturalness of the image. However, HES and NPEA achieve better performances than the proposed method. The possible reason is that these HES equalizes the illumination components rather than removing all illumination information, while NPEA was particularly designed for the naturalness preservation. Based on the analysis, we consider that our method can achieve satisfactory performance in the image naturalness preservation.

### 6.2. Frequency Domain Division Influence

In the frequency domain analysis-based feature enhancement method, the results of frequency domain division significantly influence performance of the method. As stated above, the illumination and textures of scenario are distributed in the low and high frequency, respectively. To evaluate the influence of frequency domain division and to test the sensitivity to visual image, a low-pass filter was executed repeatedly with different radius of the cutoff frequency (i.e., $D_{1}$) to acquire visual images with only low-frequency information. Five values of $D_{1}$, 5, 10, 15, 20 and 25, were used.

Figure 11 shows the results of low-pass filter on images with bright, normal and weak illumination. It can be clear that the increase of radius (i.e., $D_{1}$) leads to more brightness and textural details. The results under bright, normal and weak illuminations follow the same trend. Moreover, when $D_{1}$ became larger than 5, the textural details of the three groups of images increased without variations in illumination. As $D_{1}$ continues to increase, there are no variations in textures under the bright, normal and weak illuminations when the value of $D_{1}$ is 20, 10 or 15, respectively.

### 6.3. Parameter Influence

As stated above, the qualities of the final image optimization results strongly depend on the performance of feature enhancement. Specifically, two essential parameters can affect the result of feature enhancement in our method, which are the compression and enhancement coefficients (i.e., l,h), respectively. To evaluate the sensitivity to these two parameters, the proposed method was executed repeatedly with different pairwise sets of (l,h) on six groups of image pairs in the first dataset, which are under bright, normal and weak illuminations, respectively. Three values of l, such as 0.1, 0.2, 0.3, and six values of h in the range [0–3] were used.

The influences on the SIFT-based image matching are shown in Figure 12. The results show that NPM is maintained at a good level within a certain range of l and h. The transformed images generally achieved higher NPM than the originals, and two curves with same conditions followed similar trend. Based on the peaks of curves, the optimal values of l and h can be found to achieve best matching performances. Such as, 0.2 and 1.0 under weak and normal illumination conditions, 0.2 and 2.0 for bright illumination.

## 7. Conclusions

In this study, a spatial-frequency domain associated image-optimization method is specially designed to improve illumination-robustness image matching. An adaptive luminance equalization model is implemented in the spatial domain to equalize illumination variation instead of removing all illumination information. A frequency domain analysis-based feature-enhancement algorithm is proposed to enhance features without over-enhancement and destruction of naturalness. Our method is then used with image-matching algorithms for illumination-robust image matching. Various experiments were conducted to test the proposed approach. The results show our method achieve a higher accuracy and efficiency for image matching and a better application in SFM and MVS, compared with the other state-of-the-art methods.

The drawback of the proposed method is that the implementation is dependent on the scenario variation. In the algorithm, the division of frequency domain (i.e., $D_{1}$, $D_{2}$) and the parameters (i.e., l, h) are scene dependent, and need to be set scene adaptively. Therefore, future studies will improve the parameter setting scheme for automatic applications at high frame rates, and extend our concept to more complex scenarios. In addition, because the influence of the light source position on image intensity is complicated, we will investigate the detailed influence of the illumination variation in direction on the performances of image optimization and image matching, and take more complex illumination into account in future studies.

## Figures and Tables

**Figure 1 sensors-20-06489-f001:**
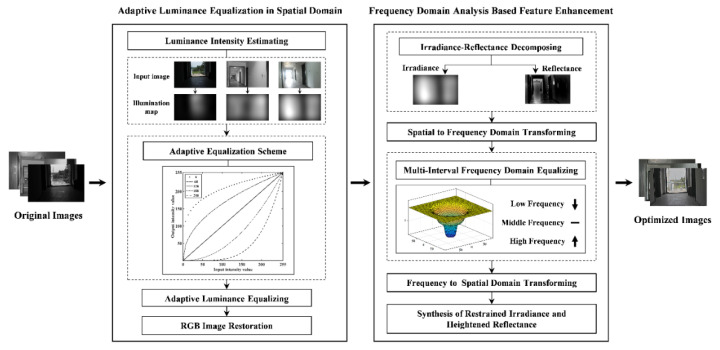
Flowchart of the proposed image-optimization method, including two steps, namely, adaptive luminance equalization and feature enhancement.

**Figure 2 sensors-20-06489-f002:**
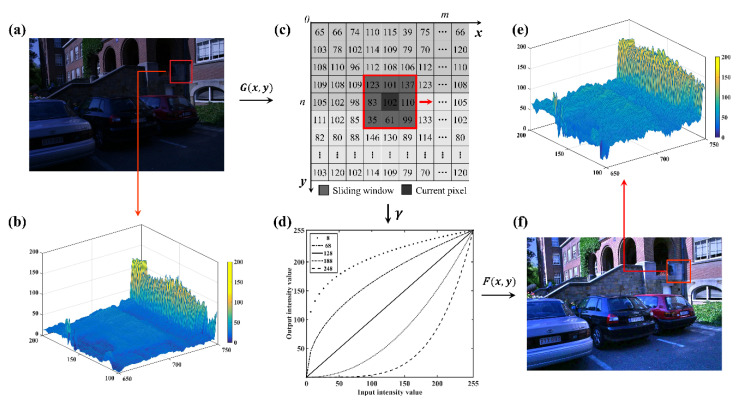
Adaptive luminance equalization: (**a**) original image *G*(*x*, *y*); (**b**) the luminance distribution of a local region for original image (red rectangle in (**a**)); (**c**) the adaptive equalizing scheme based on illumination map; (**d**) adaptive luminance equalization model; (**e**) the luminance distribution of a local region for the transformed image (red rectangle in (**f**)); and (**f**) the transformed image *F*(*x*, *y*).

**Figure 3 sensors-20-06489-f003:**
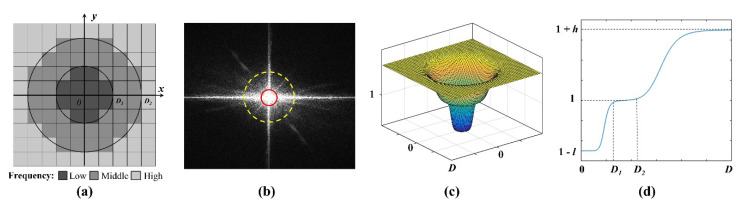
A graphical illustration of frequency domain equalizing: (**a**) the division of frequency domain; (**b**) an example of FT-based frequency power spectrum and frequency domain division (the low and high frequency region are displayed within the red and yellow circles, respectively); (**c**) multi-interval frequency domain equalization model visualized in3D; and (**d**) the multi-interval frequency domain equalization model and its performance in theory.

**Figure 4 sensors-20-06489-f004:**
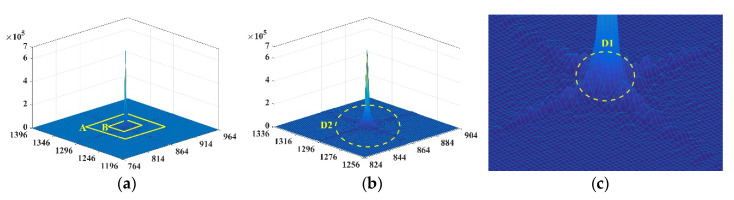
The frequency domain amplitude spectrum visualized in 3D: (**a**) the amplitude distribution of the local regions in the frequency domain; (**b**) a partial enlarged image of region A in (**a**) (the high-frequency region is depicted beyond the yellow ellipse $D_{2}$); and (**c**) a partial enlarged image of region B in (**a**) (the low-frequency region is displayed within the yellow ellipse $D_{1}$).

**Figure 5 sensors-20-06489-f005:**
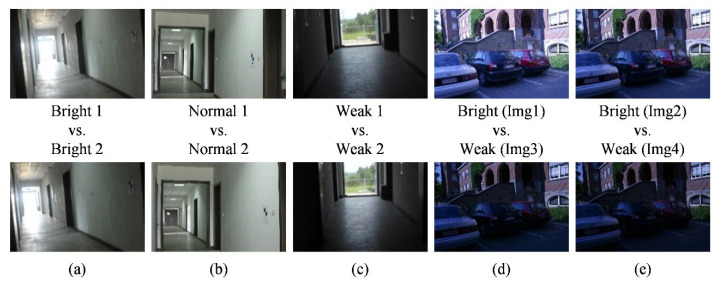
The experimental images from the datasets along with their description: (**a**) the first image pair; (**b**) the second image pair; (**c**) the third image pair; (**d**) the fourth image pair; (**e**) the fifth image pair.

**Figure 6 sensors-20-06489-f006:**
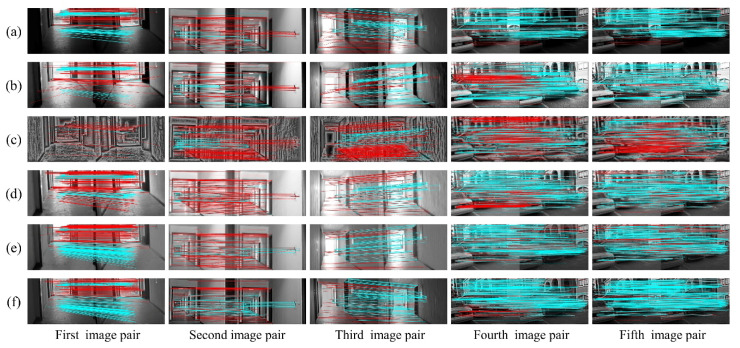
The visual matching performances when applying SIFT: (**a**) original images; (**b**) images transformed with HES; (**c**) images transformed with LFA; (**d**) images transformed with LIME; (**e**) images transformed with NPEA; and (**f**) images transformed with our method.

**Figure 7 sensors-20-06489-f007:**
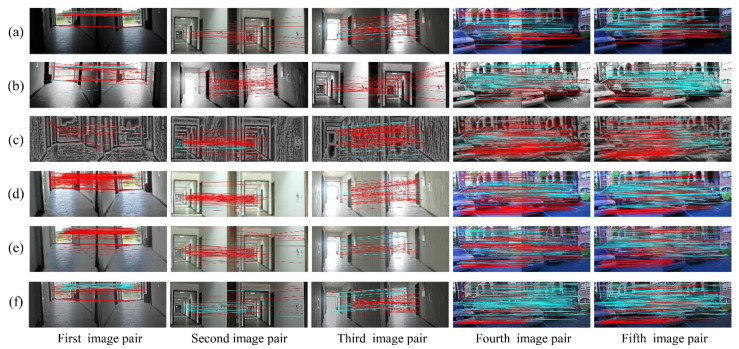
The visual matching performances when applying SURF: (**a**) original images; (**b**) images transformed with HES; (**c**) images transformed with LFA; (**d**) images transformed with LIME; (**e**) images transformed with NPEA; and (**f**) images transformed with our method.

**Figure 8 sensors-20-06489-f008:**
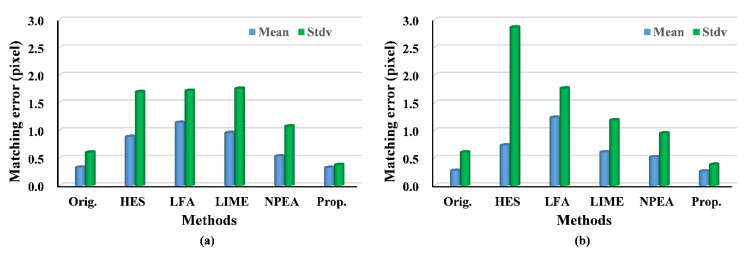
Histograms of matching error using SIFT: (**a**) the fourth image pair; (**b**) the fifth image pair.

**Figure 9 sensors-20-06489-f009:**
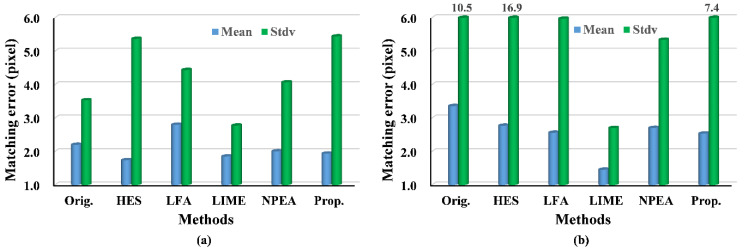
Histograms of matching error using SURF: (**a**) the fourth image pair; (**b**) the fifth image pair.

**Figure 10 sensors-20-06489-f010:**
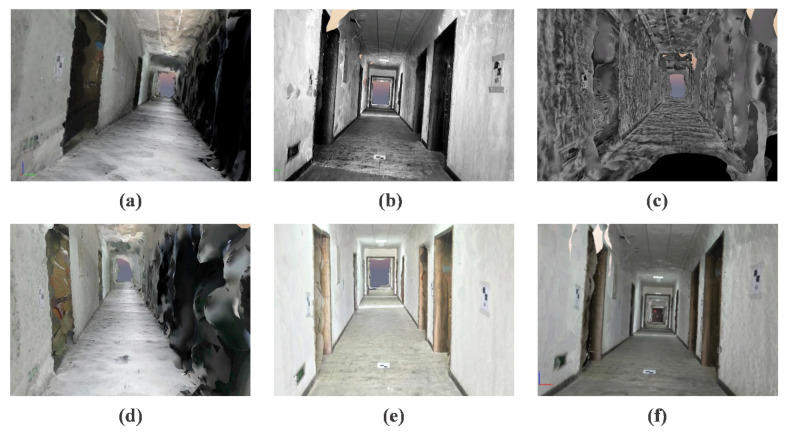
Comparison of the results of MVS with original and optimized images: (**a**) original; (**b**) HES; (**c**) LFA; (**d**) LIME; (**e**) NPEA; and (**f**) our method.

**Figure 11 sensors-20-06489-f011:**
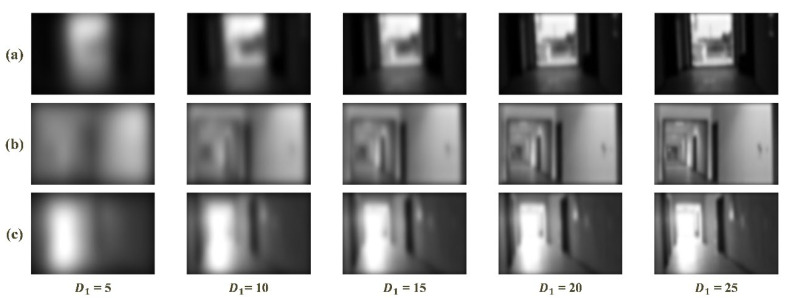
Visual results of low-pass filter with a sequence of $D_{1}$: (**a**) weak illumination; (**b**) normal illumination; and (**c**) bright illumination.

**Figure 12 sensors-20-06489-f012:**
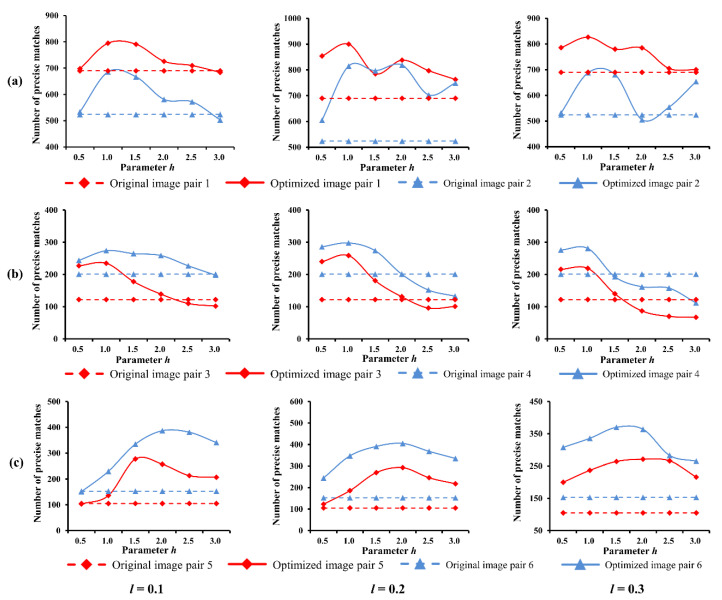
Parameter influence on image matching: (**a**) weak illumination; (**b**) normal illumination; (**c**) bright illumination.

**Table 1 sensors-20-06489-t001:** Parameter setting in this paper.

Parameter	Weak Image	Normal Image	Bright Image
Low frequency range $D_{1}$	5	5	5
High frequency range $D_{2}$	15	10	20
Compression coefficients l	0.2	0.2	0.2
Enhancement coefficients h	1	1	2

**Table 2 sensors-20-06489-t002:** Numerical performance indicators of SIFT for the first to third image pairs.

Measure	The First Image Pair (Weak)						The Second Image Pair (Normal)						The Third Image Pair (Bright)					
	Original	HES	LFA	LIME	NPEA	OURS	Original	HES	LFA	LIME	NPEA	OURS	Original	HES	LFA	LIME	NPEA	OURS
F1	6677	16,661	44,408	17,245	11,480	5930	1018	6862	53,830	6241	3477	1297	641	7073	26,274	1294	790	909
F2	7474	16,252	27,128	17,483	13,224	6422	1165	7365	55,946	6943	3793	1455	465	7278	28,232	2011	1221	730
M	215	172	42	465	385	282	135	161	331	230	156	192	118	143	321	168	105	136
NPM	140	88	8	**256**	220	189	45	77	**125**	73	60	111	64	93	**120**	93	68	100
T(s)	22.44	39.09	103.44	44.39	35.67	**20.73**	15.45	22.28	252.25	21.72	16.78	**15.11**	15.16	22.54	71.04	15.78	14.41	**14.28**
RPMF	0.021	0.005	0.000	0.014	0.019	**0.032**	0.044	0.011	0.002	0.012	0.017	**0.086**	**0.138**	0.013	0.005	0.072	0.086	**0.137**
MP	0.651	0.512	0.190	0.551	0.571	**0.670**	0.333	0.478	0.378	0.317	0.385	**0.578**	0.542	0.650	0.374	0.554	0.648	**0.735**
RPMT	6.239	2.251	0.077	5.767	6.168	**9.117**	2.913	3.456	0.496	3.361	3.576	**7.346**	4.222	4.126	1.689	5.894	4.719	**7.003**

**Table 3 sensors-20-06489-t003:** Numerical performance indicators of SURF for the first to third image pairs.

Measure	The First Image Pair (Weak)						The Second Image Pair (Normal)						The Third Image Pair (Bright)					
	Original	HES	LFA	LIME	NPEA	OURS	Original	HES	LFA	LIME	NPEA	OURS	Original	HES	LFA	LIME	NPEA	OURS
F1	4408	9738	24,377	9092	7426	6115	1761	8155	23,750	5342	3711	2179	1134	8152	19,392	2220	1721	1796
F2	4568	11,349	18,486	10,156	7787	5604	2189	8502	24,274	5415	4187	3094	1375	7969	20,080	2887	2138	2322
M	68	79	32	175	147	102	31	49	85	73	76	60	62	49	113	53	25	81
NPM	33	29	5	20	30	**59**	4	7	18	9	14	**32**	16	8	26	5	7	**34**
T(s)	16.88	33.78	96.51	31.91	25.15	**20.15**	11.91	25.49	118.6	18.94	15.72	**11.87**	10.58	24.3	82.89	13.31	12.21	**10.51**
RPMF	0.007	0.003	0.000	0.002	0.004	**0.011**	0.002	0.001	0.001	0.002	0.004	**0.015**	0.014	0.001	0.001	0.002	0.004	**0.019**
MP	0.485	0.367	0.156	0.114	0.204	**0.578**	0.129	0.143	0.212	0.123	0.184	**0.533**	0.258	0.163	0.230	0.094	0.280	**0.420**
RPMT	1.955	0.858	0.052	0.627	1.193	**2.928**	0.336	0.275	0.152	0.475	0.891	**2.696**	1.512	0.329	0.314	0.376	0.573	**3.235**

**Table 4 sensors-20-06489-t004:** Numerical performance indicators of SIFT for the fourth and fifth image pairs.

Measure	The Fourth Image Pair						The Fifth Image Pair					
	Original	HES	LFA	LIME	NPEA	OURS	Original	HES	LFA	LIME	NPEA	OURS
F1	2701	6823	2902	3576	2720	4120	1251	6730	2755	3598	2801	3636
F2	1540	6991	2870	3979	3302	3784	2314	7163	2891	3950	2950	3719
M	248	705	747	662	570	571	316	437	832	710	647	592
NPM	209	519	493	508	485	**511**	282	373	547	**594**	539	**557**
T(s)	3.47	9.88	4.27	5.21	4.24	5.15	**3.81**	9.78	4.18	5.57	4.26	5.54
RPMF	0.136	0.076	0.172	0.142	**0.178**	0.135	**0.225**	0.055	0.199	0.165	0.192	0.153
MP	0.843	0.736	0.660	0.767	0.851	**0.895**	0.892	0.854	0.657	0.837	0.833	**0.941**
RPMT	60.231	52.530	115.457	97.505	114.387	99.223	74.016	38.139	**130.861**	106.643	126.526	100.542

**Table 5 sensors-20-06489-t005:** Numerical performance indicators of SURF for the fourth and fifth image pairs.

Measure	The Fourth Image Pair						The Fifth Image Pair					
	Original	HES	LFA	LIME	NPEA	OURS	Original	HES	LFA	LIME	NPEA	OURS
F1	1762	2567	2071	2065	1725	2486	1483	2598	2066	1986	1783	2371
F2	931	2742	2170	2262	1982	2297	751	2706	2153	2220	1883	2161
M	207	315	478	454	395	428	245	344	597	508	417	485
NPM	114	213	223	227	208	**308**	137	229	331	323	246	**337**
T(s)	2.76	5.15	3.98	4.65	3.61	**2.65**	2.57	5.07	4.03	3.94	3.69	**2.55**
RPMF	0.122	0.083	0.108	0.110	0.121	**0.134**	0.182	0.088	0.160	0.163	0.138	**0.156**
MP	0.551	0.676	0.467	0.500	0.527	**0.720**	0.559	0.666	0.554	0.636	0.590	**0.695**
RPMT	41.304	41.359	56.030	48.817	57.618	**116.226**	54.582	45.168	82.134	81.980	66.667	**132.157**

**Table 6 sensors-20-06489-t006:** Performance indicators in SFM.

Indicators	Original	HES	LFA	LIME	NPEA	OURS
SRIL	69.4%	71.2%	90.0%	83.4%	78.6%	85.4%
RE	0.78	0.77	1.04	0.99	0.95	0.75

**Table 7 sensors-20-06489-t007:** Total error of checkpoints (mm).

Check Point	Illumination	Original	HES	LFA	LIME	NPEA	OURS
1	Normal	20	37	21	loss	31	17
2	Normal	36	32	19	29	39	29
3	Normal	28	31	26	49	15	25
4	Bright	22	loss	7	24	14	10
5	Bright	23	loss	39	loss	38	21
6	Weak	27	loss	62	loss	42	20
Mean		26	33	29	34	30	20

**Table 8 sensors-20-06489-t008:** Quantitative measurement results of LOE.

Images	Orig.	HES	LFA	NPEA	LIME	Prop.
Bright 1	0	**131.93**	1621.00	167.92	320.10	174.54
Bright 2	0	**131.49**	1531.00	210.94	303.74	194.57
Normal 1	0	142.22	1456.00	**123.33**	338.60	244.25
Normal 2	0	152.59	1358.00	**100.62**	318.45	262.50
Weak 1	0	**158.57**	1879.00	364.47	266.37	205.89
Weak 2	0	**154.15**	1856.00	468.04	316.28	221.72
Img1	0	546.42	821.06	**193.25**	412.88	266.66
Img2	0	523.86	847.37	**291.50**	407.24	320.27
Img3	0	550.91	903.15	555.59	404.51	**401.31**
Img4	0	545.04	920.05	650.41	422.71	**406.44**

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
