# Peer review of "A Spatial-Frequency Domain Associated Image-Optimization Method for Illumination-Robust Image Matching"

_sensors, 2020, doi:10.3390/s20226489_

Round 1

Reviewer 1 Report

This paper is well written with a sufficient explanation of their conclusions.

The authors proposed a spatial-frequency domain associated image optimization method for complex illumination image-based matching. The experimental results show the evidence of the improvement of the proposed method compared with the SOTA methods.

Here are some small concerns and suggestions:

  1. The comparison results with SIFT and SURF showed in Fig.8 and 9 are difficult for the audience to understand. I suggest the author combine subfigures into a large one. No need to show the details of the histogram of each image, just show the average error of each pair and merge the results into a figure.
  2. Why choose SIFT and SURF for the matching comparison? These two methods usually use the gray-scale image as input for feature extraction, the impact of the brightness changes on the matching results is not significant.

Reviewer 2 Report

The paper proposed an image optimization method based on spatial-frequency operations. Authors show that the method is robust to complex illumination variation, while keeping naturalness of the image. The paper is well written, and it is easy to understand, however I have several questions about the experimental results.

  1. In the Discussion Section and the feature work in the Conclusion, authors mentioned that the performance of the proposed method strongly depends on some parameters, such as diameters of circular division in frequency domain and parameters (l and h) of the frequency domain equalization model given by (11). Does it mean that the optimal values of those parameters were used, depending on image feature, to get all results in the Experimental Results Section?

If yes, please add the parameter’s values used for each image.

  1. Besides these critical parameters, the proposed method uses several parameters, such as R and c in eq. (11). R means whole image or local region of the image?
  2. Histograms shown in Figs. 8 and 9 are very difficult to compare, because the range of the y-axis (Frequency) of each figure is different. Please unify this range.
  3. PPMT means “percentage of precise matches to the processing time”. However, some values of the tables 1-4 are more than 100. How to calculate this value? Or what does it mean?  Which is unit of these values?
  4. In tables 1-4, there are F1 and F2. I suppose that these are number of features. Which is different between F1 and F2? F1 is number of features in the first image and F2 is for the second one? Please add some additional explanations
  5. What “after1”, “after2”, “before1” and “before2” of the Fig.12?

  1. In the section 6.1, “D0” means “D1” in ew. (11) , Fig. 3(a) and line 275.

  1. What means “img1-5” and img”2-6” in title of figures in Figs. 8 and 9?

  1. I do not understand well the red circle if Fig. 10, because in the case of (c), the reconstruction of whole image is failed, not only within red circle.

Additionally, I found some errors.

  • The number of tables in text is Roman number, but in table caption is Arabic.
  • Some references are incomplete. For example, 19, 28, 29, 32, etc.

Reviewer 3 Report

Dear authors thank you for your contribution. Despite the fact that your approach is sensitive to image content and parameters selection, which sound like an ad-hoc solution, it is a valid contribution to the community.

If you could add pseudocode on paragraph 4, it would helkp future scholars to replicate your method. Also it would be highly advantageous if you could release your code or executable (github or private page), so that other may test is or use it. This would raise your citations.

The two datasets you are using are representative of the kind of applications the algorithm may encounter. Nevertheless, the evaluation of the performace of such image enhancment methods, is vague. Can you please include the literature that you followed for such testing?

Also tables 1-2-3-4, could be in portrait mode. It would make them a bit smaller in width, and the paper more compact. Please add on the measures a comment like 'higher is better' or 'lower is better'. It would make it much more readable to readers unfamiliar with all measures.

Reviewer 4 Report

Brief summary

The authors introduce an image optimization technique, which improves correspondence analysis, in particular feature-based matching, in images taken under extreme and varying illumination conditions. The proposed image optimization method consists of two stages: luminance equalization, which is performed in the spatial domain, and feature enhancement performed as filtering in the frequency domain. A key concept of this work is the preservation of image naturalness and the avoidance of over-enhancement.

The authors present a detailed evaluation of their method against four other recent image enhancement algorithms. Based on their evaluation, the authors conclude that the proposed method performs better than the other methods by exhibiting higher matching precision and lower reprojection errors. The authors only point out a single drawback of their method, namely the sensitivity of their optimization process to the parameter selection. Therefore, the improvement of the parameter selection process is suggested as important future work.

General comments

The following strengths of the manuscript should be mentioned:

+ The manuscript addresses a relevant and challenging research problem.

+ The manuscript is well-structured and overall clearly written.

+ The authors have conducted a detailed evaluation incorporating several quality measures and two different test datasets.

+ Additionally, the discussion on parameter influence in section 6.2 maintains information transparency, and gives the reader a better overview over the effect of the relevant parameters.

The following aspects of the work should be improved:

(Note the comments below include the same reference numbers and abbreviations of algorithm names as those used by the authors of the manuscript)

- Additional grammar and spell checks are necessary. The manuscript contains several punctuation, grammar, and spelling mistakes, which make reading and understanding the text slightly difficult. For particular suggestions, please refer to the “specific comments” section.

- Some additional references could increase the information density of the manuscript. Specific suggestions of such references are included in the last section.

- It would be helpful for readers to include an explicit reference to the standard dataset from Oxford University used in the evaluation process.

- I could not find any information about the origin of implementations of the four comparison image enhancement algorithms. Where did the used implementation come from? Did you reimplement the algorithms? In any case, there needs to be some sort of comment or reference about the origin.

- The proposed method seems to be specifically designed to preserve image naturalness. However, the evaluation process does not include any formal assessment of the level of naturalness after the proposed method has been applied. Although image naturalness can be considered as a subjective quality, one of the four comparison methods (NPEA, reference 44), proposes such a measure. 

- Furthermore, given that the proposed image optimization method aims to improve image matching results, the connection between image naturalness and the image matching success rate remains unclear. Including references on this subject would be helpful.

- The choice of the 4 competing methods, used for the evaluation, is not explicitly motivated. This raises concerns of potentially unfair or sub-optimal comparison conditions, which further puts the validity of the presented evaluation results in question. For example, the LFA method (reference 18) is designed to improve the appearance of under-exposed images, i.e. images taken under low lighting conditions. Yet, you evaluate the LFA method also on over-exposed images.

As a second example, one might ask why you decided to compare their method exactly against the HES (reference 37) algorithm, while mentioning also other, potentially superior, histogram equalization-based methods in section 2.2. 

- The authors have considered different types of complex illumination conditions including very high or very low light intensity for both images of an image pair, as well as light intensity differences between the two images. However, it might be interesting to consider also more complex scenarios, in which the direction of the illumination changes between the images to be matched. Such a real-world scenario of image matching under varying illumination directions, for example, is described in the paper referenced under number 26.

- Finally, while it is praiseworthy that the manuscript also includes evaluation of sparse and dense matching based on the optimize images, the input dataset is reported to contain predominantly planar surfaces (walls, ceilings, floors). The same experiments could be further extended to include images of more complex geometry. As an example, the Middlebury stereo vision reference datasets provide image pairs under varying lighting conditions and varying scene complexity, in addition to ground through disparity data, which can be used to measure the reprojection errors.  

Specific comments

- Examples of different language related problems include:

  • punctuation (missing comma on line 39; a comma too much on line 75)
  • spelling mistake (line 471) ?
  • improper usage of definite articles (lines 63 and 338)
  • inconsistent title capitalization (line 87)
  • unclear sentence formulation (lines 43 and 184)
  • incorrect verb conjugation (lines 75 and 101)

- Certain claims seem to be formulated vaguely, using non-quantifiable expressions such as: ‘appropriately improved’ (line 92) or ‘few variations’ (line 527)

- Improvement suggestions related to figures and tables:

  • The changing viewing perspective of the SFM-reconstructions presented in Figure 10. should be replaced by a single viewing position, consistent among all 6 cases.
  • The description of Figure 3.b), with respect to the location of the high frequencies, seems to be incorrect or ambiguous.
  • There seems to be no explanation why certain values in Table 6. are not available.
  • There are inconsistencies between the table labels used in the text (Roman numerals) and in the table captions themselves (Arabic numerals)
  • Furthermore, there are slight inconsistencies between the parameter notation between text description and image labels in Figure 11 (D0 becomes D).

- Suggestions for additional references:

        For the definition of the concept of image naturalness:

  • Yendrikhovskij, S. & Blommaert, F. & Ridder, Huib. (1999). Color reproduction and the naturalness constraint. Color Research & Application. 24. 52 - 67. 10.1002/(SICI)1520-6378(199902)24:1<52::AID-COL10>3.0.CO;2-4.

    Recent survey papers about image matching and color constancy which would add to the related work section:
  • Ma, Jiayi & Jiang, Xingyu & Fan, Aoxiang & Jiang, Junjun & Yan, Junchi. (2020). Image Matching from Handcrafted to Deep Features: A Survey. International Journal of Computer Vision. 10.1007/s11263-020-01359-2.

  • Gijsenij, Arjan & Gevers, T. & Weijer, Joost. (2011). Computational Color Constancy: Survey and Experiments. IEEE transactions on image processing: a publication of the IEEE Signal Processing Society. 20. 2475-89. 10.1109/TIP.2011.2118224.

Round 2

Reviewer 2 Report

Authors attended satisfactory to all suggestions that I provided for the first version, so I think that the quality of the revised version is high enough for its publication.